# Lymphedema and Trismus after Head and Neck Cancer, and the Impact on Body Image and Quality of Life

**DOI:** 10.3390/cancers16030653

**Published:** 2024-02-03

**Authors:** Coralie R. Arends, Lisette van der Molen, Josephine E. Lindhout, Karoline Bragante, Arash Navran, Michiel W. M. van den Brekel, Martijn M. Stuiver

**Affiliations:** 1Department of Head and Neck Oncology and Surgery, Netherlands Cancer Institute, 1066 CX Amsterdam, The Netherlands; l.vd.molen@nki.nl (L.v.d.M.); m.vd.brekel@nki.nl (M.W.M.v.d.B.); m.stuiver@nki.nl (M.M.S.); 2Amsterdam Center for Language and Communication, University of Amsterdam, 1012 WP Amsterdam, The Netherlands; 3Faculty of Health, Medicine and Life Sciences, Maastricht University, 6211 LK Maastricht, The Netherlands; 4Department of Physical Rehabilitation, Federal University of Health Science of Porto Alegre, Porto Alegre 90050-170, Brazil; kcbragante@unisinos.br; 5Department of Radiation Oncology, Netherlands Cancer Institute, 1066 CX Amsterdam, The Netherlands; a.navran@nki.nl; 6Department of Oral and Maxillofacial Surgery, Amsterdam University Medical Center, 1081 HV Amsterdam, The Netherlands; 7Center for Quality of Life and Division of Psychosocial Research and Epidemiology, Netherlands Cancer Institute, 1066 CX Amsterdam, The Netherlands; 8Center of Expertise Urban Vitality, Faculty of Health, Amsterdam University of Applied Sciences, 1091 GC Amsterdam, The Netherlands

**Keywords:** head and neck cancer, lymphedema, trismus, quality of life, body image

## Abstract

**Simple Summary:**

After treatment for head and neck cancer, long-term sequelae can occur, such as external lymphedema and trismus. The prevalence of lymphedema varies from 35 to 75%. The aim of our cross-sectional study was to assess the prevalence of lymphedema and trismus. There were 59 patients included in the study, treated between six months to three years ago. The prevalence of lymphedema was 94.1% with a median severity score of 9 (range 0–24). Trismus was only in 1.2% present. Body image and QoL was generally good in our population. Patients with higher lymphedema scores have poorer speech with a moderate correlation.

**Abstract:**

Background: To assess the prevalence of chronic lymphedema and trismus in patients > 6 months after head and neck cancer (HNC) treatment, and to explore how the severity of these conditions correlates with body image and quality of life. Methods: The cross-sectional sample included 59 patients, treated for HNC between six months to three years ago. Physical measurements were performed to assess the presence of external lymphedema and trismus (<36 mm). Furthermore, participants completed two questionnaires regarding body image (BIS) and quality of life (UW-QoL V4). Results: Lymphedema prevalence was 94.1% (95% CI 0.86–0.98), with a median severity score of 9 (range 0–24). Trismus prevalence in this sample was 1.2%. The median BIS score was 2, indicating a positive body image. The UW-QoL score showed a good QOL with a median of 100. Only the domain of saliva and overall related health had a lower median of 70 and 60, respectively. There was no correlation between lymphedema and body image (r = 0.08, *p* = 0.544). Patients with higher lymphedema scores reported poorer speech with a moderate correlation (r = −0.39, *p* = 0.003). Conclusion: Lymphedema is a highly prevalent, but moderately severe late side-effect of HNC with a limited impact on quality of life domains except for speech, in our cohort.

## 1. Introduction

Improved and multimodal treatment options have increased the number of head and neck cancer (HNC) survivors. Unfortunately, treatment of HNC can come with significant toxicity leading to long-term sequelae, including lymphedema and trismus (restricted mouth opening) [1,2,3].

Lymphedema results from damage to the soft tissues, including the lymphatic system, as a result of surgery and radiotherapy with or without concomitant chemotherapy. Disruption of the lymphatic system due to treatment-induced tissue damage results in an overload of protein-rich interstitial fluid. When lymphedema persists for >3 months after completion of treatment, it is considered a chronic condition and referred to as secondary lymphedema [4,5]. The reported prevalence of chronic secondary lymphedema after HNC treatment varies from 35 to 75% [1,6,7]. Patients with external head and neck lymphedema (HNL) may experience sensations such as tightness, numbness, heaviness, and warmth [8]. Apart from (sub)cutaneous lymphedema, also (sub)mucosal lymphedema can occur, leading to vocal changes, swallowing problems, and sometimes airway obstruction.

Trismus is usually defined as a maximal interincisal mouth opening of 35 mm or less [7,8]. Trismus can result from tumor infiltration, radiation-induced fibrosis, or scarring after surgery of the masticatory musculature. The reported prevalence of trismus following radiotherapy for HNC ranged from 25 to 42% in previous studies, and is related to the radiation dose on the masticulatory muscles as well as the surgical procedure [9,10]. Trismus can cause impairments in many aspects of daily life, such as chewing, eating, oral hygiene, kissing and speech, and can also cause pain. Limitations in these everyday activities likely affect the quality of life (QoL) [9].

Although HNL and trismus are common late effects, research on the impact on QoL of HNL and trismus in HNC survivors is still scarce [11]. An association with impaired QoL has been reported for lymphedema in two previous studies, both of which reported a decreased QoL in patients with HNL [11,12]. Only one study related trismus to QoL and showed that the presence of trismus negatively impacts the health-related QoL and that patients showed more signs of depression [9].

The wide range in occurrence of lymphedema and trismus as reported in previous studies may be related to the variation in the anatomical sites assessed for lymphedema (external vs. internal), the difference in timing of measurement during of follow-up, different measurement instruments, difference in previous cancer treatment and differences in supportive care. In our institute, preventive exercises to limit or prevent swallowing problems and trismus have been the standard of care since 2008. This study aims to determine the prevalence of external lymphedema and trismus in a cross-sectional sample of patients with head and neck cancer treated at our institute between 6 months and 3 years ago. Additionally, if possible, we explored how the severity of these conditions correlates with body image and domains of quality of life.

## 2. Materials and Methods

### 2.1. Study Design and Participants

We performed a cross-sectional study in which we collected data of patients treated for HNC with (chemo)radiotherapy, with or without surgery, at the Netherlands Cancer Institute. A convenience sample was collected during two periods: 20 October 2018 to 20 January 2019 and 15 January 2020 to 15 July 2020. Patients who finished their oncological treatment between six months to three years ago and >18 years were eligible for inclusion. Patients were excluded when there was evidence of recurrent disease or when they had been treated with palliative intent.

All patients with a scheduled appointment at the Netherlands Cancer Institute (NKI) during the study period were contacted by phone at least one week before the appointment, to ask if they wanted to participate in the study. All participants provided written informed consent before participation in this study. The study was approved by the NKI Institutional Review Board (IRBd20-027).

### 2.2. Measurements

We collected patient characteristics including sex, age (at the day of inclusion), and body mass index (BMI: kg/m^2^); tumor characteristics (localization of the primary tumor and TNM classification [13]); and treatment details (chemotherapy, surgery, and radiotherapy).

Lymphedema was measured by rating pitting edema at a total of 10 locations (all bilateral) in the head and neck. Locations were based on the head and neck external lymphedema-fibrosis assessment criteria version 2 [14]. The grading of the pitting edema was performed on a 4-point scale that is often used to assess the severity of HNL [15]. HNL severity is based on the pit’s depth and the duration in seconds it takes for the skin to revert to its original status. Digital pressure with the thumb was applied for 20 s per area. Each side was scored with the 4-point scale at each of the five measuring points per side mentioned above, resulting in a total HNL score ranging between 0 and 40. A higher score indicates more severe HNL [14,15].

Mouth opening was measured to quantify trismus. Active range of motion (ROM) for mouth opening was measured using the Therabite ROM scale (Atos Medical AB). The edge of the millimeter ruler was placed between the lower central incisors, and vertically measured the space between the upper central incisors. Mouth opening can be reliably assessed with the Therabite ROM scale [16]. To further enhance reliability, two measurements were taken, and their mean was recorded as the final value [16,17]. Trismus was defined using the threshold of 35 mm (mm) or less, based on the study of Dijkstra et al. [18].

The Body Image Scale (BIS) is a brief patient reported outcome measure for assessing body image changes in patients with cancer developed by the EORTC [19]. A validated Dutch translation is available, which was shown to be reliable among cancer patients [20]. The BIS consists of 10 questions with a 4-point scale, scored 0–3. The total score ranges from 0 (minimum body image-related distress) to 30 (maximum distress) [19,20,21].

The University of Washington Quality of Life questionnaire V4 (UW-Qol V4) a health-related quality of life questionnaire [22]. It is a commonly used measurement tool for head and neck cancer patients and consists of 12 domains. These domains are pain, appearance, activity, recreation, swallowing, chewing, speech, shoulder, taste, saliva, mood, and anxiety. Each question has three to six response options and is scaled between a 0 (worst) to 100 (best) score according to the response hierarchy. There is one final question where patients choose the three domains that matter most to them. Furthermore, there are also three final questions on global quality of life questions. With a panel of clinical experts, we decided to focus on the domains swallowing, chewing, speech, saliva, pain, and the question on overall health-related QoL over the last seven days, since these domains were deemed likely to be affected by trismus and HNL.

### 2.3. Statistical Analysis

Descriptive statistics were used to summarize sample characteristics and study outcomes, with Wilson 95% confidence intervals (95% CI) for the prevalence estimates. Continuous data were summarized using mean and standard deviations, or medians and quartiles in case of clear deviation from a normal distribution. Categorical variables were summarized as numbers and percentages.

The correlations were calculated using Spearman’s correlation coefficient. We considered a correlation lower than 0.39 as weak, between 0.39 and 0.60 as moderate, between 0.60 and 0.80 as strong, and greater than 0.80 as very strong [23]. Statistical significance was assumed at a *p*-value < 0.05 for all tests. All analyses were performed with R software version 4.0.2 for MacOS.

## 3. Results

### 3.1. Patient Characteristics

In total, 148 eligible patients visited the outpatient clinic during the two recruitment periods. Of these, 103 patients were successfully approached, and 59 patients agreed to participate. The mean (SD) age of participants was 60.6 (10) years, and 69.5% were male. The oropharynx was the most common (32%) primary tumor site. There was a large variation in tumor stages. One patient had been diagnosed with distant metastasis but was treated with curative intent, nonetheless. Aside from radiotherapy, 20.3% of patients had also been treated with chemotherapy and 7% with immunotherapy. One in four had been treated with a selective or comprehensive neck dissection. Full patient characteristics are shown in Table 1. Except for missing the TNM classification for one participant, there was no missing data. Compared to participants, there were not notable differences in received treatment, with exception of a higher percentage of concomitant chemoradiotherapy among the non-participants (43 vs. 22%).

### 3.2. Lymphedema and Trismus

The median total HNL score was 9, with a range of 0 to 24. The overall prevalence of HNL, i.e., a score > 0, was 94.1% (95% CI: 0.86–0.98). In total, 64% had a maximum score ≤ 2 on any location. Figure 1 shows the total pitting edema score (PES) and the score per side. The average mouth opening of the two measurements was 47.5 mm, with a range of 31 to 70 mm. Only 2 patients (1.8%, 95% CI: 0–0.11) had trismus as defined by a mouth opening of 35 mm or less.

### 3.3. Body Image and Quality of Life

Overall, most included patients reported a positive body image on the BIS (median 2, IQR 1 to 5). Only two participants (3.4%) reported impaired body image with scores of 19 and 26, respectively. Thirteen patients (22%) had the most optimal score for body image.

In the UW-QoL V4 scores, the domain saliva had a median of 70, and the overall health-related question a median of 60, as can be seen in Figure 2. The domains swallowing, chewing, speech, and pain (69.5%) all had a median of 100, with 41 (69.5%), 48 (81.4%), and 34 (57.6%) of the patients achieving the highest possible score.

### 3.4. Correlation

The correlations of HNL scores with body image and QoL domains of interest are listed in Table 2. There was no significant correlation between HNL and body image (r = 0.08) or any of the QoL domains, except for a moderate correlation (−0.39, *p* = 0.003) with the domain speech.

## 4. Discussion

This study aimed to estimate the prevalence of HNL and trismus in HNC survivors treated >6 months ago. In addition, we explored whether the severity of these conditions correlate with patients’ body image and selected quality of life domains. The prevalence of HNL in this sample was high; 94.1%. However, HNL was mild in most patients. Correlations of HNL with body image and quality of life domains were weak and non-significant, except for a moderate correlation between HNL and the domain speech (rho = −0.39).

The prevalence of HNL in our study was higher than that reported in the literature [7]. Deng et al. (2012) described a prevalence of HNL in 75% [7]. This difference in prevalence is most likely largely due to our sensitive criterion for HNL presence (pitting score ≥ 1). Another explanation could be related to the treatment modality. All patients in our sample had received radiotherapy, as compared to 90% in the study performed by Deng et al. Different prevalences in the literature could also be related to timing of measurement relative to HNC treatment. In the study by Deng et al., the meantime post-treatment was 24.71 months, whereas in our study this was 12.9 months. This could suggest that HNL prevalence may decrease over time in this population. Indeed, a post hoc analysis showed that HNL severity in our sample declined over time.

Only two patients (both treated for a T4 tumor) in our study sample (1.8%) had a mouth opening <35 mm. A previous systematic review reported an association of time since treatment with trismus prevalence, with a peak in prevalence of 44.1% at six months after treatment [24]. According to the same study, tumor locations in the oral cavity or oropharynx were associated with a higher risk of developing trismus compared to hypopharynx or larynx cancers, and treatment with conventional radiotherapy was identified as a risk factor compared to intensity-modulated radiation therapy (IMRT). Based on the association with tumor location, a substantially higher trismus prevalence would be expected in our sample, since 44% had oral cavity or oropharynx cancer. A possible explanation for the low trismus prevalence is that all our patients had received IMRT. Better results after IMRT (compared to conventional RT) were reported in a systematic review by Bensadoun et al. (2010). They found a trismus prevalence of >25% after conventional RT or with chemotherapy, compared to 5% after IMRT (5%) [25]. The latter result is comparable with the prevalence of trismus in our study. In addition to the standard use of IMRT, an integrated rehabilitation program is standard care at our institute. This includes exercises aimed at trismus prevention for all patients at risk. There is evidence that such exercises can prevent trismus, as we have reported previously [3,26,27,28,29]. The low incidence of trismus prohibited useful analyses of the correlation between trismus and body image or quality of life. Future studies could selectively include patients with trismus to shed more light on these issues.

Contrary to a previous study, we found no significant correlation between HNL scores and self-reported swallowing, saliva and chewing, nor with pain or self-reported overall health [12,30]. The body image and QoL (UW-QoL) scores were very high in our population and consequently the scales showed a ceiling effect. Therefore, lack of correlation could be attributed, in part, by a lack of variation on body image and quality of life scores. Aside from the general lack of correlation, in a post hoc exploration of the raw data, the two patients who reported body image impairments (BIS scores of 26 and 19) had HNL scores of only 5 and 13, respectively. It is conceivable that patients with HNL value their symptoms differently compared to patients with lymphedema resulting from (treatment of) other cancers. Future qualitative research could provide better insight into patient experiences of HNL.

### Limitations

A limitation of this study is the relatively small sample size. The sample size calculation was based on a higher anticipated prevalence, for which a sample size of 50 patients would have been sufficient. The lower prevalence prohibited correlational analyses with trismus. Other limitations include the cross-sectional design and the possibility of selective-non response. On average, non-responders had more advanced tumors and there were more patients treated with combined chemo-radiotherapy, but the mean dose of radiation was lower than in the included patient population. On the other hand, considering the objectives of the study, it would be reasonable to assume that selective non-response would be related to low levels of complaints. Our sample exhibited low trismus prevalence, good quality of life, and mild HNL scores. Thus, we believe non-response related to study outcomes is not very likely or may be so negligible that a large influence on the outcome is unlikely. Also, although we used convenience sampling in two cross-sectional time periods, the findings in both periods were comparable, which further strengthens our belief that the results are representative of our population. Although we included patients at different points in time since treatment, the cross-sectional nature of the study prohibited direct analysis of change in trismus and HNL prevalence over time. Another limitation is that HNL measurement was limited to severity of pitting edema, and did not include other aspects such as patient experience of HNL symptoms or the presence of fibrosis. It is possible that we misclassified patients with later stages of HNL who had low levels of pitting as a result of fibrosis. For future research, it is desirable to examine changes over time in a prospective cohort design, so the course of HNL within individual patients can be examined. Such a study could also shed light on risk factors for persistent HNL.

## 5. Conclusions

In this cross-sectional study in a specialized cancer hospital, we observed a high prevalence of mostly mild pitting HNL and a low prevalence of trismus 6 months to 3 years after treatment including radiotherapy for HNC. Severity of pitting HNL was not associated with body image or quality of life domains, with exception of speech. Although HNL is a frequent side-effect of HNC, the impact on the quality of life seems limited as long as HNL is mild, but qualitative research is needed to better understand its impact on individual patients.

## Figures and Tables

**Figure 1 cancers-16-00653-f001:**
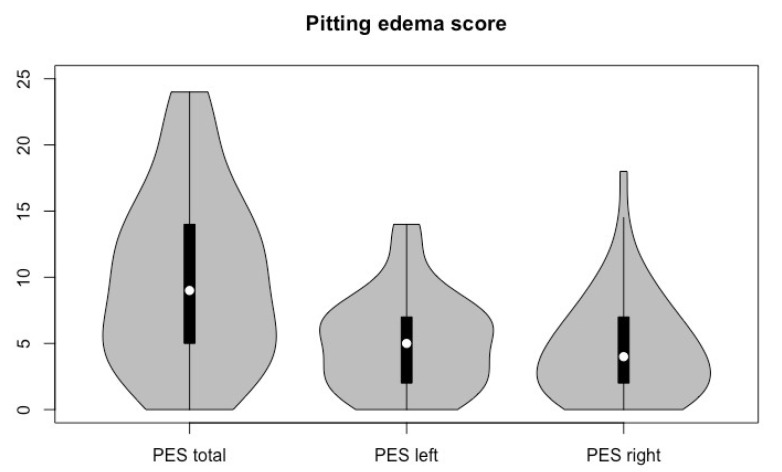
Violin plot of pitting lymphedema scores.

**Figure 2 cancers-16-00653-f002:**
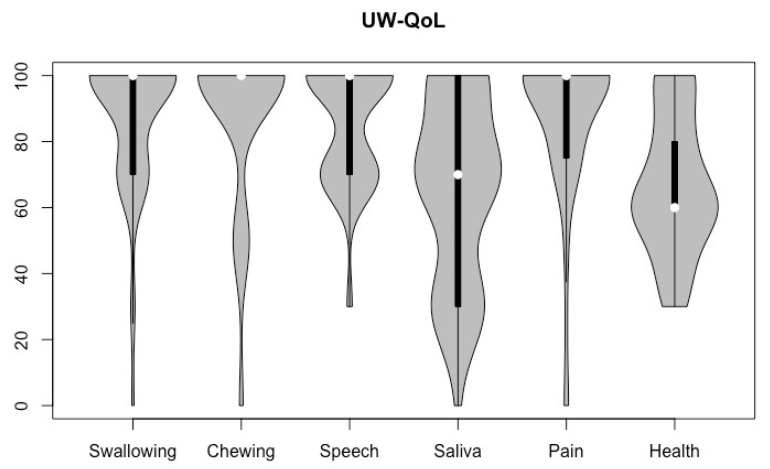
Violin plot of UW-QoL V4 domains of all participants (higher scores indicate better QoL).

**Table 1 cancers-16-00653-t001:** Patient characteristics.

Characteristics	(*N* = 59)
Sex (Male)	41 (69.5%)
Age	60.6 (10) ^1^
BMI	24.3 (22–25.9) ^2^
Time since treatment (months)	12.9 (3.3) ^1^
*Primary site*	
Skin	4 (6.8%)
Hypopharynx	3 (5.1%)
Larynx	7 (11.9%)
Oral cavity	7 (11.9%)
Nasopharynx	5 (8.5%)
Oropharynx	19 (32%)
Thyroid	1 (1.7%)
Salivary glad	7 (11.9%)
Other	3 (5.1%)
Unknown primary	3 (5.1%)
*T-class*	
X	6 (10.2%)
1	18 (30.5%)
2	14 (23.7%)
3	7 (11.9%)
4	13 (22%)
NA	1 (1.7%)
*N-class*	
0	21 (35.6%)
1	17 (28.8%)
2	16 (27.1%)
3	4 (6.8%)
NA	1 (1.7%)
*M-class*	
0	58 (96.6%)
1	1 (1.7%)
NA	1 (1.7%)
*Treatment received*	
Radiotherapy	
Concomitant	13 (22%)
Postoperative	22 (37.3%)
Primary	24 (40.7%)
Chemotherapy (concomintant Cisplatin)	12 (20.3%)
Surgery (selevtive/comprehensive neck dissection)	
No	44 (74.6%)
Unilateral	13 (22%)
Bilateral	2 (3.4%)
Immunotherapy	4 (7%)
Avelumab	1 (1.7%)
Ipilimumab	3 (5.1%)
Nivolumab	3 (5.1%)

N; number of patients, BMI; Body Mass Index, ^1^ mean (standard deviation), ^2^ median (Inter quartile range).

**Table 2 cancers-16-00653-t002:** Correlation HNL with body image (BIS, range 0 to 30) and QoL domains (UW-QoL V4, range 0–100).

Correlates	Spearman rho	95% CI	*p*-Value
BIS	0.08	−0.19–0.35	0.544
Swallowing	0.02	−0.26–0.26	0.868
Chewing	−0.19	−0.46–0.1	0.143
Speech	−0.39	−0.61–−0.14	0.003
Saliva	0.03	−0.21–0.31	0.830
Pain	0.06	−0.24–0.32	0.639
Overall health	−0.09	−0.24–0.32	0.516

## Data Availability

Research data are stored in an institutional repository and can be shared upon request to the corresponding author and after ethical clearance of the NKI Institutional Review Board.

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
