# Peer review of "Lymphedema and Trismus after Head and Neck Cancer, and the Impact on Body Image and Quality of Life"

_cancers, 2024, doi:10.3390/cancers16030653_

Round 1
Reviewer 1 Report (Previous Reviewer 1)
Comments and Suggestions for Authors
Thank you for letting me review this, I'm happy with the changes and the replies to my comments.
Comments on the Quality of English Language
A very small linguistic comment on pages 120 and 201 a point is missing after et al (it should be et al.)
Reviewer 2 Report (Previous Reviewer 2)
Comments and Suggestions for Authors
Lymphedema and trismus are common issues for head and neck patients.
It is a very nice study; I congratulate you on the paper; it was a real pleasure reviewing it.
Thank you
Reviewer 3 Report (Previous Reviewer 3)
Comments and Suggestions for Authors
In most papers the QoL is higher reduced. Maybe because the edema wasn't as severe as we see it. But the paper shows interesting results in trismus that haven't shown before.
This manuscript is a resubmission of an earlier submission. The following is a list of the peer review reports and author responses from that submission.
Round 1
Reviewer 1 Report
Comments and Suggestions for Authors
Thank you for letting me review this. Interesting and important topic.
I only have a few suggestions.
TNM classification reference is missing.
Why not use cancer stage as well?
Mouth opening was measured with the Therabite ROM scale, at each time was it only measured once, some suggest that it should be measured three times to get a correct value?
Most patients are T- 0,1 and 2 are there any differences between T 3 and 4 (or Stage) in the prevalence of lymphedema and trismus?
There is a rather small sample size that needs to be discussed in the limitation section.
Comments on the Quality of English LanguageAbbreviations should be used correctly throughout the whole manuscript (except for purpose/aim when it is not needed) as it is now it is not e.g. line 70 and 189 HNC, 72 and 237 QOL.
Reviewer 2 Report
Comments and Suggestions for Authors
It is a very interesting subject, the design is good but the COnclusion area must be improved.
Comments on the Quality of English Languagesmall corrections are nedd it
Reviewer 3 Report
Comments and Suggestions for Authors
I see big differences in your results to other authors: Late side effects of radiation treatment for head and neck cancer, I. Brook, 2020; Lymphoedema after head and neck cancer treatment: an overview for clinical practice, C. Jeans, B. Brown, E. C. Ward and A. E. Vertigan, 2021;
What does mild lymphedema mean? There is the possibility of measuring the head lymphedema not only by pitting. I#m astonished to see that the lymphedema of head and face doesn't influence the QoL so much
